# Semi-Supervised Learning with Ladder Networks

**Antti Rasmus and Harri Valpola**
The Curious AI Company, Finland

**Mikko Honkala**
Nokia Labs, Finland

**Mathias Berglund and Tapani Raiko**
Aalto University, Finland & The Curious AI Company, Finland

## Abstract

We combine supervised learning with unsupervised learning in deep neural networks. The proposed model is trained to simultaneously minimize the sum of supervised and unsupervised cost functions by backpropagation, avoiding the need for layer-wise pre-training. Our work builds on top of the Ladder network proposed by Valpola [1] which we extend by combining the model with supervision. We show that the resulting model reaches state-of-the-art performance in semi-supervised MNIST and CIFAR-10 classification in addition to permutation-invariant MNIST classification with all labels.

## 1 Introduction

In this paper, we introduce an unsupervised learning method that fits well with supervised learning. Combining an auxiliary task to help train a neural network was proposed by Suddarth and Kergosien [2]. There are multiple choices for the unsupervised task, for example reconstruction of the inputs at every level of the model [e.g., 3] or classification of each input sample into its own class [4].

Although some methods have been able to simultaneously apply both supervised and unsupervised learning [3, 5], often these unsupervised auxiliary tasks are only applied as pre-training, followed by normal supervised learning [e.g., 6]. In complex tasks there is often much more structure in the inputs than can be represented, and unsupervised learning cannot, by definition, know what will be useful for the task at hand. Consider, for instance, the autoencoder approach applied to natural images: an auxiliary decoder network tries to reconstruct the original input from the internal representation. The autoencoder will try to preserve all the details needed for reconstructing the image at pixel level, even though classification is typically invariant to all kinds of transformations which do not preserve pixel values.

Our approach follows Valpola [1] who proposed a Ladder network where the auxiliary task is to denoise representations at every level of the model. The model structure is an autoencoder with skip connections from the encoder to decoder and the learning task is similar to that in denoising autoencoders but applied at every layer, not just the inputs. The skip connections relieve the pressure to represent details at the higher layers of the model because, through the skip connections, the decoder can recover any details discarded by the encoder. Previously the Ladder network has only been demonstrated in unsupervised learning [1, 7] but we now combine it with supervised learning.

The key aspects of the approach are as follows:

**Compatibility with supervised methods**. The unsupervised part focuses on relevant details found by supervised learning. Furthermore, it can be added to existing feedforward neural networks, for example multi-layer perceptrons (MLPs) or convolutional neural networks (CNNs).

**Scalability due to local learning**. In addition to supervised learning target at the top layer, the model has local unsupervised learning targets on every layer making it suitable for very deep neural networks. We demonstrate this with two deep supervised network architectures.

**Computational efficiency**. The encoder part of the model corresponds to normal supervised learning. Adding a decoder, as proposed in this paper, approximately triples the computation during training but not necessarily the training time since the same result can be achieved faster due to better utilization of available information. Overall, computation per update scales similarly to whichever supervised learning approach is used, with a small multiplicative factor.

As explained in Section 2, the skip connections and layer-wise unsupervised targets effectively turn autoencoders into hierarchical latent variable models which are known to be well suited for semi-supervised learning. Indeed, we obtain state-of-the-art results in semi-supervised learning in the MNIST, permutation invariant MNIST and CIFAR-10 classification tasks (Section 4). However, the improvements are not limited to semi-supervised settings: for the permutation invariant MNIST task, we also achieve a new record with the normal full-labeled setting.For a longer version of this paper with more complete descriptions, please see [8].

## 2 Derivation and justification

Latent variable models are an attractive approach to semi-supervised learning because they can combine supervised and unsupervised learning in a principled way. The only difference is whether the class labels are observed or not. This approach was taken, for instance, by Goodfellow et al. [5] with their multi-prediction deep Boltzmann machine. A particularly attractive property of hierarchical latent variable models is that they can, in general, leave the details for the lower levels to represent, allowing higher levels to focus on more invariant, abstract features that turn out to be relevant for the task at hand.

The training process of latent variable models can typically be split into inference and learning, that is, finding the posterior probability of the unobserved latent variables and then updating the underlying probability model to better fit the observations. For instance, in the expectation-maximization (EM) algorithm, the E-step corresponds to finding the expectation of the latent variables over the posterior distribution assuming the model fixed and M-step then maximizes the underlying probability model assuming the expectation fixed.

The main problem with latent variable models is how to make inference and learning efficient. Suppose there are layers $l$ of latent variables $\mathbf{z}^{(l)}$. Latent variable models often represent the probability distribution of all the variables explicitly as a product of terms, such as $p(\mathbf{z}^{(l)} \mid \mathbf{z}^{(l+1)})$ in directed graphical models. The inference process and model updates are then derived from Bayes' rule, typically as some kind of approximation. Often the inference is iterative as it is generally impossible to solve the resulting equations in a closed form as a function of the observed variables.

There is a close connection between denoising and probabilistic modeling. On the one hand, given a probabilistic model, you can compute the optimal denoising. Say you want to reconstruct a latent $z$ using a prior $p(z)$ and an observation $\tilde{z} = z + \text{noise}$. We first compute the posterior distribution $p(z \mid \tilde{z})$, and use its center of gravity as the reconstruction $\hat{z}$. One can show that this minimizes the expected denoising cost $(\hat{z} - z)^2$. On the other hand, given a denoising function, one can draw samples from the corresponding distribution by creating a Markov chain that alternates between corruption and denoising [9].

Valpola [1] proposed the Ladder network where the inference process itself can be learned by using the principle of denoising which has been used in supervised learning [10], denoising autoencoders (dAE) [11] and denoising source separation (DSS) [12] for complementary tasks. In dAE, an autoencoder is trained to reconstruct the original observation $\mathbf{x}$ from a corrupted version $\tilde{\mathbf{x}}$. Learning is based simply on minimizing the norm of the difference of the original $\mathbf{x}$ and its reconstruction $\hat{\mathbf{x}}$ from the corrupted $\tilde{\mathbf{x}}$, that is the cost is $\|\hat{\mathbf{x}} - \mathbf{x}\|^2$.

While dAEs are normally only trained to denoise the observations, the DSS framework is based on the idea of using denoising functions $\hat{\mathbf{z}} = g(\mathbf{z})$ of latent variables $\mathbf{z}$ to train a mapping $\mathbf{z} = f(\mathbf{x})$ which models the likelihood of the latent variables as a function of the observations. The cost function is identical to that used in a dAE except that latent variables $\mathbf{z}$ replace the observations $\mathbf{x}$,

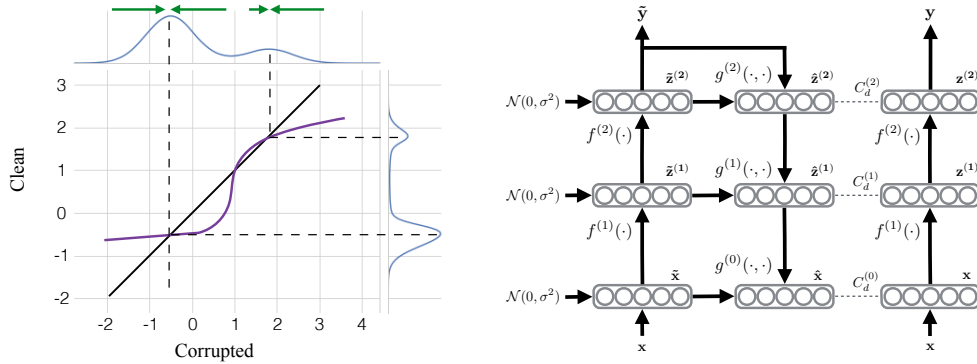

Figure 1: **Left**: A depiction of an optimal denoising function for a bimodal distribution. The input for the function is the corrupted value (x axis) and the target is the clean value (y axis). The denoising function moves values towards higher probabilities as show by the green arrows. **Right**: A conceptual illustration of the Ladder network when $L = 2$. The feedforward path ($\mathbf{x} \to \mathbf{z}^{(1)} \to \mathbf{z}^{(2)} \to \mathbf{y}$) shares the mappings $f^{(l)}$ with the corrupted feedforward path, or encoder ($\mathbf{x} \to \tilde{\mathbf{z}}^{(1)} \to \tilde{\mathbf{z}}^{(2)} \to \tilde{\mathbf{y}}$). The decoder ($\tilde{\mathbf{z}}^{(l)} \to \hat{\mathbf{z}}^{(l)} \to \hat{\mathbf{x}}$) consists of denoising functions $g^{(l)}$ and has cost functions $C_d^{(l)}$ on each layer trying to minimize the difference between $\hat{\mathbf{z}}^{(l)}$ and $\mathbf{z}^{(l)}$. The output $\tilde{\mathbf{y}}$ of the encoder can also be trained to match available labels $t(n)$.

that is, the cost is $\|\hat{\mathbf{z}} - \mathbf{z}\|^2$. The only thing to keep in mind is that $\mathbf{z}$ needs to be normalized somehow as otherwise the model has a trivial solution at $\mathbf{z} = \hat{\mathbf{z}} = $ constant. In a dAE, this cannot happen as the model cannot change the input $\mathbf{x}$.

Figure 1 (left) depicts the optimal denoising function $\hat{z} = g(\tilde{z})$ for a one-dimensional bimodal distribution which could be the distribution of a latent variable inside a larger model. The shape of the denoising function depends on the distribution of $z$ and the properties of the corruption noise. With no noise at all, the optimal denoising function would be the identity function. In general, the denoising function pushes the values towards higher probabilities as shown by the green arrows.

Figure 1 (right) shows the structure of the Ladder network. Every layer contributes to the cost function a term $C_d^{(l)} = \|\mathbf{z}^{(l)} - \hat{\mathbf{z}}^{(l)}\|^2$ which trains the layers above (both encoder and decoder) to learn the denoising function $\hat{\mathbf{z}}^{(l)} = g^{(l)}(\tilde{\mathbf{z}}^{(l)}, \hat{\mathbf{z}}^{(l+1)})$ which maps the corrupted $\tilde{\mathbf{z}}^{(l)}$ onto the denoised estimate $\hat{\mathbf{z}}^{(l)}$. As the estimate $\hat{\mathbf{z}}^{(l)}$ incorporates all prior knowledge about $\mathbf{z}$, the same cost function term also trains the encoder layers below to find cleaner features which better match the prior expectation.

Since the cost function needs both the clean $\mathbf{z}^{(l)}$ and corrupted $\tilde{\mathbf{z}}^{(l)}$, during training the encoder is run twice: a clean pass for $\mathbf{z}^{(l)}$ and a corrupted pass for $\tilde{\mathbf{z}}^{(l)}$. Another feature which differentiates the Ladder network from regular dAEs is that each layer has a skip connection between the encoder and decoder. This feature mimics the inference structure of latent variable models and makes it possible for the higher levels of the network to leave some of the details for lower levels to represent. Rasmus et al. [7] showed that such skip connections allow dAEs to focus on abstract invariant features on the higher levels, making the Ladder network a good fit with supervised learning that can select which information is relevant for the task at hand.

One way to picture the Ladder network is to consider it as a collection of nested denoising autoencoders which share parts of the denoising machinery between each other. From the viewpoint of the autoencoder at layer $l$, the representations on the higher layers can be treated as hidden neurons. In other words, there is no particular reason why $\hat{\mathbf{z}}^{(l+i)}$ produced by the decoder should resemble the corresponding representations $\mathbf{z}^{(l+i)}$ produced by the encoder. It is only the cost function $C_d^{(l+i)}$ that ties these together and forces the inference to proceed in a reverse order in the decoder. This sharing helps a deep denoising autoencoder to learn the denoising process as it splits the task into meaningful sub-tasks of denoising intermediate representations.

**Algorithm 1** Calculation of the output $\mathbf{y}$ and cost function $C$ of the Ladder network

**Require:** $\mathbf{x}(n)$
  # Corrupted encoder and classifier
  $\tilde{\mathbf{h}}^{(0)} \leftarrow \tilde{\mathbf{z}}^{(0)} \leftarrow \mathbf{x}(n) + \mathtt{noise}$
  **for** l = 1 **to** L **do**
    $\tilde{\mathbf{z}}^{(l)} \leftarrow \mathtt{batchnorm}(\mathbf{W}^{(l)}\tilde{\mathbf{h}}^{(l-1)}) + \mathtt{noise}$
    $\tilde{\mathbf{h}}^{(l)} \leftarrow \mathtt{activation}(\boldsymbol{\gamma}^{(l)} \odot (\tilde{\mathbf{z}}^{(l)} + \boldsymbol{\beta}^{(l)}))$
  **end for**
  $P(\tilde{\mathbf{y}} \mid \mathbf{x}) \leftarrow \tilde{\mathbf{h}}^{(L)}$
  # Clean encoder (for denoising targets)
  $\mathbf{h}^{(0)} \leftarrow \mathbf{z}^{(0)} \leftarrow \mathbf{x}(n)$
  **for** l = 1 **to** L **do**
    $\mathbf{z}_{\mathrm{pre}}^{(l)} \leftarrow \mathbf{W}^{(l)}\mathbf{h}^{(l-1)}$
    $\boldsymbol{\mu}^{(l)} \leftarrow \mathtt{batchmean}(\mathbf{z}_{\mathrm{pre}}^{(l)})$
    $\boldsymbol{\sigma}^{(l)} \leftarrow \mathtt{batchstd}(\mathbf{z}_{\mathrm{pre}}^{(l)})$
    $\mathbf{z}^{(l)} \leftarrow \mathtt{batchnorm}(\mathbf{z}_{\mathrm{pre}}^{(l)})$
    $\mathbf{h}^{(l)} \leftarrow \mathtt{activation}(\boldsymbol{\gamma}^{(l)} \odot (\mathbf{z}^{(l)} + \boldsymbol{\beta}^{(l)}))$
  **end for**

# Final classification:
$P(\mathbf{y} \mid \mathbf{x}) \leftarrow \mathbf{h}^{(L)}$
# Decoder and denoising
**for** l = L **to** 0 **do**
  **if** l = L **then**
    $\mathbf{u}^{(L)} \leftarrow \mathtt{batchnorm}(\tilde{\mathbf{h}}^{(L)})$
  **else**
    $\mathbf{u}^{(l)} \leftarrow \mathtt{batchnorm}(\mathbf{V}^{(l+1)}\hat{\mathbf{z}}^{(l+1)})$
  **end if**
  $\forall i : \hat{z}_i^{(l)} \leftarrow g(\tilde{z}_i^{(l)}, u_i^{(l)})$
  $\forall i : \hat{z}_{i,\mathrm{BN}}^{(l)} \leftarrow \frac{\hat{z}_i^{(l)} - \mu_i^{(l)}}{\sigma_i^{(l)}}$
**end for**
# Cost function $C$ for training:
$C \leftarrow 0$
**if** $t(n)$ **then**
  $C \leftarrow -\log P(\tilde{\mathbf{y}} = t(n) \mid \mathbf{x}(n))$
**end if**
$C \leftarrow C + \sum_{l=0}^{L} \lambda_l \left\| \mathbf{z}^{(l)} - \hat{\mathbf{z}}_{\mathrm{BN}}^{(l)} \right\|^2$

## 3 Implementation of the Model

We implement the Ladder network for fully connected MLP networks and for convolutional networks. We used standard rectifier networks with batch normalization applied to each preactivation. The feedforward pass of the full Ladder network is listed in Algorithm 1.

In the decoder, we parametrize the denoising function such that it supports denoising of conditionally independent Gaussian latent variables, conditioned on the activations $\hat{\mathbf{z}}^{(l+1)}$ of the layer above. The denoising function $g$ is therefore coupled into components $\hat{z}_i^{(l)} = g_i(\tilde{z}_i^{(l)}, u_i^{(l)}) = \left( \tilde{z}_i^{(l)} - \mu_i(u_i^{(l)}) \right) v_i(u_i^{(l)}) + \mu_i(u_i^{(l)})$ where $u_i^{(l)}$ propagates information from $\hat{\mathbf{z}}^{(l+1)}$ by $\mathbf{u}^{(l)} = \mathtt{batchnorm}(\mathbf{V}^{(l+1)}\hat{\mathbf{z}}^{(l+1)})$. The functions $\mu_i(u_i^{(l)})$ and $v_i(u_i^{(l)})$ are modeled as expressive nonlinearities: $\mu_i(u_i^{(l)}) = a_{1,i}^{(l)} \mathtt{sigmoid}(a_{2,i}^{(l)} u_i^{(l)} + a_{3,i}^{(l)}) + a_{4,i}^{(l)} u_i^{(l)} + a_{5,i}^{(l)}$, with the form of the nonlinearity similar for $v_i(u_i^{(l)})$. The decoder has thus 10 unit-wise parameters $a$, compared to the two parameters ($\gamma$ and $\beta$ [13]) in the encoder.

It is worth noting that a simple special case of the decoder is a model where $\lambda_l = 0$ when $l < L$. This corresponds to a denoising cost only on the top layer and means that most of the decoder can be omitted. This model, which we call the $\Gamma$-model due to the shape of the graph, is useful as it can easily be plugged into any feedforward network without decoder implementation.

Further implementation details of the model can be found in the supplementary material or Ref. [8].

## 4 Experiments

We ran experiments both with the MNIST and CIFAR-10 datasets, where we attached the decoder both to fully-connected MLP networks and to convolutional neural networks. We also compared the performance of the simpler $\Gamma$-model (Sec. 3) to the full Ladder network.

With convolutional networks, our focus was exclusively on semi-supervised learning. We make claims neither about the optimality nor the statistical significance of the supervised baseline results.

We used the Adam optimization algorithm [14]. The initial learning rate was 0.002 and it was decreased linearly to zero during a final annealing phase. The minibatch size was 100. The source code for all the experiments is available at `https://github.com/arasmus/ladder`.

Table 1: A collection of previously reported MNIST test errors in the permutation invariant setting followed by the results with the Ladder network. * = SVM. Standard deviation in parenthesis.

| Test error % with # of used labels | 100 | 1000 | All |
|---|---|---|---|
| Semi-sup. Embedding [15] | 16.86 | 5.73 | 1.5 |
| Transductive SVM [from 15] | 16.81 | 5.38 | 1.40* |
| MTC [16] | 12.03 | 3.64 | 0.81 |
| Pseudo-label [17] | 10.49 | 3.46 | |
| AtlasRBF [18] | 8.10 ($\pm$ 0.95) | 3.68 ($\pm$ 0.12) | 1.31 |
| DGN [19] | 3.33 ($\pm$ 0.14) | 2.40 ($\pm$ 0.02) | 0.96 |
| DBM, Dropout [20] | | | 0.79 |
| Adversarial [21] | | | 0.78 |
| Virtual Adversarial [22] | 2.12 | 1.32 | 0.64 ($\pm$ 0.03) |
| Baseline: MLP, BN, Gaussian noise | 21.74 ($\pm$ 1.77) | 5.70 ($\pm$ 0.20) | 0.80 ($\pm$ 0.03) |
| $\Gamma$-model (Ladder with only top-level cost) | 3.06 ($\pm$ 1.44) | 1.53 ($\pm$ 0.10) | 0.78 ($\pm$ 0.03) |
| Ladder, only bottom-level cost | 1.09 ($\pm$0.32) | 0.90 ($\pm$ 0.05) | 0.59 ($\pm$ 0.03) |
| Ladder, full | **1.06** ($\pm$ 0.37) | **0.84** ($\pm$ 0.08) | **0.57** ($\pm$ 0.02) |

## 4.1 MNIST dataset

For evaluating semi-supervised learning, we randomly split the 60 000 training samples into 10 000-sample validation set and used $M = 50\,000$ samples as the training set. From the training set, we randomly chose $N = 100$, 1000, or all labels for the supervised cost.[1] All the samples were used for the decoder which does not need the labels. The validation set was used for evaluating the model structure and hyperparameters. We also balanced the classes to ensure that no particular class was over-represented. We repeated the training 10 times varying the random seed for the splits.

After optimizing the hyperparameters, we performed the final test runs using all the $M = 60\,000$ training samples with 10 different random initializations of the weight matrices and data splits. We trained all the models for 100 epochs followed by 50 epochs of annealing.

### 4.1.1 Fully-connected MLP

A useful test for general learning algorithms is the permutation invariant MNIST classification task. We chose the layer sizes of the baseline model to be 784-1000-500-250-250-250-10.

The hyperparameters we tuned for each model are the noise level that is added to the inputs and to each layer, and denoising cost multipliers $\lambda^{(l)}$. We also ran the supervised baseline model with various noise levels. For models with just one cost multiplier, we optimized them with a search grid $\{\ldots, 0.1, 0.2, 0.5, 1, 2, 5, 10, \ldots\}$. Ladder networks with a cost function on all layers have a much larger search space and we explored it much more sparsely. For the complete set of selected denoising cost multipliers and other hyperparameters, please refer to the code.

The results presented in Table 1 show that the proposed method outperforms all the previously reported results. Encouraged by the good results, we also tested with $N = 50$ labels and got a test error of 1.62 % ($\pm$ 0.65 %).

The simple $\Gamma$-model also performed surprisingly well, particularly for $N = 1000$ labels. With $N = 100$ labels, all models sometimes failed to converge properly. With bottom level or full cost in Ladder, around 5 % of runs result in a test error of over 2 %. In order to be able to estimate the average test error reliably in the presence of such random outliers, we ran 40 instead of 10 test runs with random initializations.

Table 2: CNN results for MNIST

| Test error without data augmentation % with # of used labels | 100 | all |
|---|---|---|
| EmbedCNN [15] | 7.75 | |
| SWWAE [24] | 9.17 | 0.71 |
| Baseline: Conv-Small, supervised only | 6.43 ($\pm$ 0.84) | 0.36 |
| Conv-FC | 0.99 ($\pm$ 0.15) | |
| Conv-Small, $\Gamma$-model | **0.89** ($\pm$ 0.50) | |

### 4.1.2 Convolutional networks

We tested two convolutional networks for the general MNIST classification task and focused on the 100-label case. The first network was a straight-forward extension of the fully-connected network tested in the permutation invariant case. We turned the first fully connected layer into a convolution with 26-by-26 filters, resulting in a 3-by-3 spatial map of 1000 features. Each of the 9 spatial locations was processed independently by a network with the same structure as in the previous section, finally resulting in a 3-by-3 spatial map of 10 features. These were pooled with a global mean-pooling layer. We used the same hyperparameters that were optimal for the permutation invariant task. In Table 2, this model is referred to as Conv-FC.

With the second network, which was inspired by ConvPool-CNN-C from Springenberg et al. [23], we only tested the $\Gamma$-model. The exact architecture of this network is detailed in the supplementary material or Ref. [8]. It is referred to as Conv-Small since it is a smaller version of the network used for CIFAR-10 dataset.

The results in Table 2 confirm that even the single convolution on the bottom level improves the results over the fully connected network. More convolutions improve the $\Gamma$-model significantly although the variance is still high. The Ladder network with denoising targets on every level converges much more reliably. Taken together, these results suggest that combining the generalization ability of convolutional networks[2] and efficient unsupervised learning of the full Ladder network would have resulted in even better performance but this was left for future work.

## 4.2 Convolutional networks on CIFAR-10

The CIFAR-10 dataset consists of small 32-by-32 RGB images from 10 classes. There are 50 000 labeled samples for training and 10 000 for testing. We decided to test the simple $\Gamma$-model with the convolutional architecture ConvPool-CNN-C by Springenberg et al. [23]. The main differences to ConvPool-CNN-C are the use of Gaussian noise instead of dropout and the convolutional per-channel batch normalization following Ioffe and Szegedy [25]. For a more detailed description of the model, please refer to model Conv-Large in the supplementary material.

The hyperparameters (noise level, denoising cost multipliers and number of epochs) for all models were optimized using $M = 40\,000$ samples for training and the remaining $10\,000$ samples for validation. After the best hyperparameters were selected, the final model was trained with these settings on all the $M = 50\,000$ samples. All experiments were run with with 4 different random initializations of the weight matrices and data splits. We applied global contrast normalization and whitening following Goodfellow et al. [26], but no data augmentation was used.

The results are shown in Table 3. The supervised reference was obtained with a model closer to the original ConvPool-CNN-C in the sense that dropout rather than additive Gaussian noise was used for regularization.[3] We spent some time in tuning the regularization of our fully supervised baseline model for $N = 4\,000$ labels and indeed, its results exceed the previous state of the art. This tuning was important to make sure that the improvement offered by the denoising target of the $\Gamma$-model is

Table 3: Test results for CNN on CIFAR-10 dataset without data augmentation

| Test error % with # of used labels | 4 000 | All |
|---|---|---|
| All-Convolutional ConvPool-CNN-C [23] | | 9.31 |
| Spike-and-Slab Sparse Coding [27] | 31.9 | |
| Baseline: Conv-Large, supervised only | 23.33 ($\pm$ 0.61) | 9.27 |
| Conv-Large, $\Gamma$-model | **20.40** ($\pm$ 0.47) | |

not a sign of poorly regularized baseline model. Although the improvement is not as dramatic as with MNIST experiments, it came with a very simple addition to standard supervised training.

## 5  Related Work

Early works in semi-supervised learning [28, 29] proposed an approach where inputs $\mathbf{x}$ are first assigned to clusters, and each cluster has its class label. Unlabeled data would affect the shapes and sizes of the clusters, and thus alter the classification result. Label propagation methods [30] estimate $P(y \mid \mathbf{x})$, but adjust probabilistic labels $q(y(n))$ based on the assumption that nearest neighbors are likely to have the same label. Weston et al. [15] explored deep versions of label propagation.

There is an interesting connection between our $\Gamma$-model and the contractive cost used by Rifai et al. [16]: a linear denoising function $\hat{z}_i^{(L)} = a_i \tilde{z}_i^{(L)} + b_i$, where $a_i$ and $b_i$ are parameters, turns the denoising cost into a stochastic estimate of the contractive cost. In other words, our $\Gamma$-model seems to combine clustering and label propagation with regularization by contractive cost.

Recently Miyato et al. [22] achieved impressive results with a regularization method that is similar to the idea of contractive cost. They required the output of the network to change as little as possible close to the input samples. As this requires no labels, they were able to use unlabeled samples for regularization.

The Multi-prediction deep Boltzmann machine (MP-DBM) [5] is a way to train a DBM with back-propagation through variational inference. The targets of the inference include both supervised targets (classification) and unsupervised targets (reconstruction of missing inputs) that are used in training simultaneously. The connections through the inference network are somewhat analogous to our lateral connections. Specifically, there are inference paths from observed inputs to reconstructed inputs that do not go all the way up to the highest layers. Compared to our approach, MP-DBM requires an iterative inference with some initialization for the hidden activations, whereas in our case, the inference is a simple single-pass feedforward procedure.

Kingma et al. [19] proposed deep generative models for semi-supervised learning, based on variational autoencoders. Their models can be trained with the variational EM algorithm, stochastic gradient variational Bayes, or stochastic backpropagation. Compared with the Ladder network, an interesting point is that the variational autoencoder computes the posterior estimate of the latent variables with the encoder alone while the Ladder network uses the decoder too to compute an implicit posterior approximate (the encoder provides the likelihood part which gets combined with the prior).

Zeiler et al. [31] train deep convolutional autoencoders in a manner comparable to ours. They define max-pooling operations in the encoder to feed the max function upwards to the next layer, while the argmax function is fed laterally to the decoder. The network is trained one layer at a time using a cost function that includes a pixel-level reconstruction error, and a regularization term to promote sparsity. Zhao et al. [24] use a similar structure and call it the stacked what-where autoencoder (SWWAE). Their network is trained simultaneously to minimize a combination of the supervised cost and reconstruction errors on each level, just like ours.

## 6  Discussion

We showed how a simultaneous unsupervised learning task improves CNN and MLP networks reaching the state-of-the-art in various semi-supervised learning tasks. Particularly the performance

obtained with very small numbers of labels is much better than previous published results which shows that the method is capable of making good use of unsupervised learning. However, the same model also achieves state-of-the-art results and a significant improvement over the baseline model with full labels in permutation invariant MNIST classification which suggests that the unsupervised task does not disturb supervised learning.

The proposed model is simple and easy to implement with many existing feedforward architectures, as the training is based on backpropagation from a simple cost function. It is quick to train and the convergence is fast, thanks to batch normalization.

Not surprisingly, the largest improvements in performance were observed in models which have a large number of parameters relative to the number of available labeled samples. With CIFAR-10, we started with a model which was originally developed for a fully supervised task. This has the benefit of building on existing experience but it may well be that the best results will be obtained with models which have far more parameters than fully supervised approaches could handle.

An obvious future line of research will therefore be to study what kind of encoders and decoders are best suited for the Ladder network. In this work, we made very little modifications to the encoders whose structure has been optimized for supervised learning and we designed the parametrization of the vertical mappings of the decoder to mirror the encoder: the flow of information is just reversed. There is nothing preventing the decoder to have a different structure than the encoder.

An interesting future line of research will be the extension of the Ladder networks to the temporal domain. While there exist datasets with millions of labeled samples for still images, it is prohibitively costly to label thousands of hours of video streams. The Ladder networks can be scaled up easily and therefore offer an attractive approach for semi-supervised learning in such large-scale problems.

## Acknowledgements

We have received comments and help from a number of colleagues who would all deserve to be mentioned but we wish to thank especially Yann LeCun, Diederik Kingma, Aaron Courville, Ian Goodfellow, Søren Sønderby, Jim Fan and Hugo Larochelle for their helpful comments and suggestions. The software for the simulations for this paper was based on Theano [32] and Blocks [33]. We also acknowledge the computational resources provided by the Aalto Science-IT project. The Academy of Finland has supported Tapani Raiko.

## Footnotes

[1] In all the experiments, we were careful not to optimize any parameters, hyperparameters, or model choices based on the results on the held-out test samples. As is customary, we used 10 000 labeled validation samples even for those settings where we only used 100 labeled samples for training. Obviously this is not something that could be done in a real case with just 100 labeled samples. However, MNIST classification is such an easy task even in the permutation invariant case that 100 labeled samples there correspond to a far greater number of labeled samples in many other datasets.

[2]In general, convolutional networks excel in the MNIST classification task. The performance of the fully supervised Conv-Small with all labels is in line with the literature and is provided as a rough reference only (only one run, no attempts to optimize, not available in the code package).

[3]Same caveats hold for this fully supervised reference result for all labels as with MNIST: only one run, no attempts to optimize, not available in the code package.

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
