[Supplementary Material]

# A  Details of the Implementation of the Model

The implementation details of the decoder for the MLP and convolutional models are presented in the following section.

## A.1  Decoder for Unsupervised Learning

When designing a suitable decoder to support unsupervised learning, we had to make a choice as to what kind of distributions of the latent variables the decoder would optimally be able to denoise. Based on preliminary analyses, we ultimately ended up choosing a parametrization that supports optimal denoising of latent variables that are independently distributed Gaussian variables conditional on the values of the latent variables of the layer above. The distribution of the latent variables $\mathbf{z}^{(l)}$ is therefore assumed to follow the distribution

$$p(\mathbf{z}^{(l)} \mid \mathbf{z}^{(l+1)}) = \prod_i p(z_i^{(l)} \mid \mathbf{z}^{(l+1)})$$

where $p(z_i^{(l)} \mid \mathbf{z}^{(l+1)})$ are conditionally independent Gaussian distributions of latent variables $z_i$ in layer $l$.

It can be shown that the functional form of $z_i^{(l)} = g_i(\tilde{z}_i^{(l)}, u_i^{(l)})$ has to be linear in order to minimize the denoising cost with the assumption that both the noise and the latent variable have a Gaussian distribution [1, Section 4.1]. The denoising function was therefore chosen to have the form

$$\hat{z}_i^{(l)} = g_i(\tilde{z}_i^{(l)}, u_i^{(l)}) = \left(\tilde{z}_i^{(l)} - \mu_i(u_i^{(l)})\right) v_i(u_i^{(l)}) + \mu_i(u_i^{(l)}) \tag{1}$$

where $u_i^{(l)}$ is a function of $\hat{\mathbf{z}}^{(l+i)}$. Somewhat arbitrarily, we choose the functional form pf $\mathbf{u}^{(l)}$ to be a vertical mapping from $\hat{\mathbf{z}}^{(l+1)}$, which is then batch normalized:

$$\mathbf{u}^{(l)} = \mathtt{batchnorm}(\mathbf{V}^{(l+1)} \hat{\mathbf{z}}^{(l+1)}),$$

where the matrix $\mathbf{V}^{(l)}$ has the same dimension as the transpose of $\mathbf{W}^{(l)}$ on the encoder side. The projection vector $\mathbf{u}^{(l)}$ therefore has the same dimensionality as $\mathbf{z}^{(l)}$. Furthermore, the functions $\mu_i(u_i^{(l)})$ and $v_i(u_i^{(l)})$ are modeled as expressive nonlinearities:

$$\mu_i(u_i^{(l)}) = a_{1,i}^{(l)} \mathtt{sigmoid}(a_{2,i}^{(l)} u_i^{(l)} + a_{3,i}^{(l)}) + a_{4,i}^{(l)} u_i^{(l)} + a_{5,i}^{(l)}$$

$$v_i(u_i^{(l)}) = a_{6,i}^{(l)} \mathtt{sigmoid}(a_{7,i}^{(l)} u_i^{(l)} + a_{8,i}^{(l)}) + a_{9,i}^{(l)} u_i^{(l)} + a_{10,i}^{(l)},$$

where $a_{1,i}^{(l)} \ldots a_{10,i}^{(l)}$ are the trainable parameters of the nonlinearity for each neuron $i$ in each layer $l$.

For the lowest layer, $\hat{\mathbf{x}} = \hat{\mathbf{z}}^{(0)}$ and $\tilde{\mathbf{x}} = \tilde{\mathbf{z}}^{(0)}$ by definition, and for the highest layer we chose $\mathbf{u}^{(L)} = \tilde{\mathbf{y}}$. This allows the highest-layer denoising function to utilize prior information about the classes being mutually exclusive which seems to improve convergence in cases where there are very few labeled samples.

One interpretation of this formulation is that we are modeling the distribution of $\mathbf{z}^{(l)}$ as a mixture of Gaussians with diagonal covariance matrices, where the value of the above layer $\mathbf{z}^{(l+1)}$ modulates the form of the Gaussian that $\mathbf{z}^{(l)}$ is distributed as. In this parametrization all correlations, non-linearities and non-Gaussianities in the latent variables $\mathbf{z}^{(l)}$ have to be represented by modulations from above layers for optimal denoising. The parametrizaiton therefore encourages the decoder to find representations $\mathbf{z}^{(l)}$ that have high mutual information with $\mathbf{z}^{(l+1)}$. This is crucial as it allows supervised learning to have an indirect influence on the representations learned by the unsupervised decoder: any abstractions selected by supervised learning will bias the lower levels to find more representations which carry information about the same abstractions.

Rasmus et al. [7] showed that modulated connections in $g$ are crucial for allowing the decoder to recover discarded details from the encoder and thus for allowing invariant representations to develop. The proposed parametrization can represent such modulation but also traditional top-down decoder connections that are normally used in dAEs. We also tested alternative formulations for the denoising function, the results of which can be found in Appendix C.

The cost function for the unsupervised path is the mean squared reconstruction error per neuron, but there is a slight twist which we found to be important. Batch normalization has useful properties, but it also introduces noise which affects both the clean and corrupted encoder pass. This noise is highly correlated between $\mathbf{z}^{(l)}$ and $\tilde{\mathbf{z}}^{(l)}$ because the noise derives from the statistics of the samples that happen to be in the same minibatch. This highly correlated noise in $\mathbf{z}^{(l)}$ and $\tilde{\mathbf{z}}^{(l)}$ biases the denoising functions to be simple copies[4] $\hat{\mathbf{z}}^{(l)} \approx \tilde{\mathbf{z}}^{(l)}$.

The solution we found was to implicitly use the projections $\mathbf{z}_{\text{pre}}^{(l)}$ as the target for denoising and scale the cost function in such a way that the term appearing in the error term is the batch normalized $\mathbf{z}^{(l)}$ instead. For the moment, let us see how that works for a scalar case:

$$\frac{1}{\sigma^2} \| z_{\text{pre}} - \hat{z} \|^2 = \left\| \frac{z_{\text{pre}} - \mu}{\sigma} - \frac{\hat{z} - \mu}{\sigma} \right\|^2 = \| z - \hat{z}_{\text{BN}} \|^2$$

$$z = \texttt{batchnorm}(z_{\text{pre}}) = \frac{z_{\text{pre}} - \mu}{\sigma}$$

$$\hat{z}_{\text{BN}} = \frac{\hat{z} - \mu}{\sigma},$$

where $\mu$ and $\sigma$ are the batch mean and batch std of $z_{\text{pre}}$, respectively, that were used in batch normalizing $z_{\text{pre}}$ into $z$. The unsupervised denoising cost function $C_{\text{d}}$ is thus

$$C_{\text{d}} = \sum_{l=0}^{L} \lambda_l C_{\text{d}}^{(l)} = \sum_{l=0}^{L} \frac{\lambda_l}{N m_l} \sum_{n=1}^{N} \left\| \mathbf{z}^{(l)}(n) - \hat{\mathbf{z}}_{\text{BN}}^{(l)}(n) \right\|^2, \tag{2}$$

where $m_l$ is the layer's width, N the number of training samples, and the hyperparameter $\lambda_l$ a layer-wise multiplier determining the importance of the denoising cost.

The model parameters $\mathbf{W}^{(l)}, \boldsymbol{\gamma}^{(l)}, \boldsymbol{\beta}^{(l)}, \mathbf{V}^{(l)}, \mathbf{a}_i^{(l)}$ can be trained simply by using the backpropagation algorithm to optimize the total cost $C = C_{\text{c}} + C_{\text{d}}$. The feedforward pass of the full Ladder network is listed in Algorithm 1. Classification results are read from the $\mathbf{y}$ in the clean feedforward path.

## A.2 Variations

Section A.1 detailed how to build a decoder for the Ladder network to match a fully connected encoder. It is easy to extend the same approach to other encoders, for instance, convolutional neural networks (CNN). For the decoder of fully connected networks we used vertical mappings whose shape is a transpose of the encoder mapping. The same treatment works for the convolution operations: in the networks we have tested in this paper, the decoder has convolutions whose parametrization mirrors the encoder and effectively just reverses the flow of information. As the idea of convolution is to reduce the number of parameters by weight sharing, we applied this to the parameters of the denoising function g, too.

Many convolutional networks use pooling operations with stride, that is, they downsample the spatial feature maps. The decoder needs to compensate this with a corresponding upsampling. There are several alternative ways to implement this and in this paper we chose the following options: 1) on the encoder side, pooling operations are treated as separate layers with their own batch normalization and linear activations function and 2) the downsampling of the pooling on the encoder side is compensated by upsampling with copying on the decoder side. This provides multiple targets for the decoder to match, helping the decoder to recover the information lost on the encoder side.

It is worth noting that a simple special case of the decoder is a model where $\lambda_l = 0$ when $l < L$. This corresponds to a denoising cost only on the top layer and means that most of the decoder can be omitted. This model, which we call the $\Gamma$-model due to the shape of the graph, is useful as it can easily be plugged into any feedforward network without decoder implementation. In addition, the $\Gamma$-model is the same for MLPs and convolutional neural networks. The encoder in the $\Gamma$-model still includes both the clean and the corrupted paths as in the full ladder.

# B  Specification of the convolutional models

Table 4: Description ConvPool-CNN-C by Springenberg et al. [23] and our networks based on it.

| Model | | |
|---|---|---|
| ConvPool-CNN-C | Conv-Large (for CIFAR-10) | Conv-Small (for MNIST) |
| Input $32 \times 32$ or $28 \times 28$ RGB or monochrome image | | |
| $3 \times 3$ conv. 96 ReLU | $3 \times 3$ conv. 96 BN LeakyReLU | $5 \times 5$ conv. 32 ReLU |
| $3 \times 3$ conv. 96 ReLU | $3 \times 3$ conv. 96 BN LeakyReLU | |
| $3 \times 3$ conv. 96 ReLU | $3 \times 3$ conv. 96 BN LeakyReLU | |
| $3 \times 3$ max-pooling stride 2 | $2 \times 2$ max-pooling stride 2 BN | $2 \times 2$ max-pooling stride 2 BN |
| $3 \times 3$ conv. 192 ReLU | $3 \times 3$ conv. 192 BN LeakyReLU | $3 \times 3$ conv. 64 BN ReLU |
| $3 \times 3$ conv. 192 ReLU | $3 \times 3$ conv. 192 BN LeakyReLU | $3 \times 3$ conv. 64 BN ReLU |
| $3 \times 3$ conv. 192 ReLU | $3 \times 3$ conv. 192 BN LeakyReLU | |
| $3 \times 3$ max-pooling stride 2 | $2 \times 2$ max-pooling stride 2 BN | $2 \times 2$ max-pooling stride 2 BN |
| $3 \times 3$ conv. 192 ReLU | $3 \times 3$ conv. 192 BN LeakyReLU | $3 \times 3$ conv. 128 BN ReLU |
| $1 \times 1$ conv. 192 ReLU | $1 \times 1$ conv. 192 BN LeakyReLU | |
| $1 \times 1$ conv. 10 ReLU | $1 \times 1$ conv. 10 BN LeakyReLU | $1 \times 1$ conv. 10 BN ReLU |
| global meanpool | global meanpool BN | global meanpool BN |
| | | fully connected 10 BN |
| 10-way softmax | | |

Here we describe two model structures, Conv-Small and Conv-Large, that were used for MNIST and CIFAR-10 datasets, respectively. They were both inspired by ConvPool-CNN-C by Springenberg et al. [23]. Table 4 details the model architectures and differences between the models in this work and ConvPool-CNN-C. It is noteworthy that this architecture does not use any fully connected layers, but replaces them with a global mean pooling layer just before the softmax function. The main differences between our models and ConvPool-CNN-C are the use of Gaussian noise instead of dropout and the convolutional per-channel batch normalization following Ioffe and Szegedy [13]. We also used 2x2 stride 2 max-pooling instead of 3x3 stride 2 max-pooling. LeakyReLU was used to speed up training, as mentioned by Springenberg et al. [23]. We utilized batch normalization in all layers, including pooling layers. Gaussian noise was also added to all layers, instead of applying dropout in only some of the layers as with ConvPool-CNN-C.

# C  Formulation of the Denoising Function

The denoising function $g$ tries to map the clean $\mathbf{z}^{(l)}$ to the reconstructed $\hat{\mathbf{z}}^{(l)}$, where $\hat{\mathbf{z}}^{(l)} = g(\tilde{\mathbf{z}}^{(l)}, \hat{\mathbf{z}}^{(l+1)})$. The reconstruction is therefore based on the corrupted value, and the reconstruction of the layer above.

An optimal functional form of $g$ depends on the conditional distribution $p(\mathbf{z}^{(l)} \mid \mathbf{z}^{(l+1)})$ that we want the model to be able to denoise. For example, if the distribution $p(\mathbf{z}^{(l)} \mid \mathbf{z}^{(l+1)})$ is Gaussian, the optimal function $g$, that is the function that achieves the lowest reconstruction error, is going to be linear with respect to $\tilde{\mathbf{z}}^{(l)}$ [1, Section 4.1]. This is the parametrization that we chose based on preliminary comparisons of different denoising function parametrizations.

The proposed parametrization of the denoising function was therefore:

$$g(\tilde{z}, u) = (\tilde{z} - \mu(u)) \, v(u) + \mu(u) \,. \tag{3}$$

We modeled both $\mu(u)$ and $v(u)$ with an expressive nonlinearity [5]: $\mu(u) = a_1 \texttt{sigmoid}(a_2 u + a_3) + a_4 u + a_5$ and $v(u) = a_6 \texttt{sigmoid}(a_7 u + a_8) + a_9 u + a_{10}$. We have left out the superscript $(l)$ and subscript $i$ in order not to clutter the equations. Given $u$, this parametrization is linear with respect to $\tilde{z}$, and both the slope and the bias depended nonlinearly on $u$.

In order to test whether the elements of the proposed function $g$ were necessary, we systematically removed components from $g$ or changed $g$ altogether and compared to the results obtained with the original parametrization. We tuned the hyperparameters of each comparison model separately

using a grid search over some of the relevant hyperparameters. However, the standard deviation of additive Gaussian corruption noise was set to 0.3. This means that the comparison do not include the best-performing models reported in Table 1 that achieved the best validation errors after more careful hyperparameter tuning.

As in the proposed function $g$, all comparison denoising functions mapped neuron-wise the corrupted hidden layer pre-activation $\tilde{\mathbf{z}}^{(l)}$ to the reconstructed hidden layer activation given one projection from the reconstruction of the layer above: $\hat{z}_i^{(l)} = g(\tilde{z}_i^{(l)}, u_i^{(l)})$.

| Test error % with # of used labels | 100 | 1000 |
|---|---|---|
| Proposed $g$: Gaussian $z$ | **1.06** ($\pm$ 0.07) | **1.03** ($\pm$ 0.06) |
| Comparison $g_1$: miniature MLP with $\tilde{z}u$ | 1.11 ($\pm$ 0.07) | 1.11 ($\pm$ 0.06) |
| Comparison $g_2$: No augmented term $\tilde{z}u$ | 2.03 ($\pm$ 0.09) | 1.70 ($\pm$ 0.08) |
| Comparison $g_3$: Linear $g$ but with $\tilde{z}u$ | 1.49 ($\pm$ 0.10) | 1.30 ($\pm$ 0.08) |
| Comparison $g_4$: Only the mean depends on $u$ | 2.90 ($\pm$ 1.19) | 2.11 ($\pm$ 0.45) |

Table 5: Semi-supervised results from the MNIST dataset. The proposed function $g$ is compared to alternative parametrizations. Note that the hyperparameter search was not as exhaustive as in the final results, which means that the results of the proposed model deviate slightly from the final results presented in Table 1

The comparison functions $g_{1\ldots4}$ are parametrized as follows:

**Comparison $g_1$: Miniature MLP with $\tilde{z}u$**

$$\hat{z} = g(\tilde{z}, u) = \mathbf{a}\boldsymbol{\xi} + b\,\texttt{sigmoid}(\mathbf{c}\boldsymbol{\xi}) \tag{4}$$

where $\xi = [1, \tilde{z}, u, \tilde{z}u]^T$ is an augmented input, $\mathbf{a}$ and $\mathbf{c}$ are trainable weight vectors, $b$ is a trainable scalar weight. This parametrization is capable of learning denoising of several different distributions including sub- and super-Gaussian and bimodal distributions.

**Comparison $g_2$: No augmented term**

$$g_2(\tilde{z}, u) = \mathbf{a}\boldsymbol{\xi}' + b\,\texttt{sigmoid}(\mathbf{c}\boldsymbol{\xi}') \tag{5}$$

where $\xi' = [1, \tilde{z}, u]^T$. $g_2$ therefore differs from $g_1$ in that the input lacks the augmented term $\tilde{z}u$.

**Comparison $g_3$: Linear g**

$$g_3(\tilde{z}, u) = \mathbf{a}\boldsymbol{\xi}. \tag{6}$$

$g_3$ differs from $g$ in that it is linear and does not have a sigmoid term. As this formulation is linear, it only supports Gaussian distributions. Although the parametrization has the augmented term that lets $u$ modulate the slope and shift of the distribution, the scope of possible denoising functions is still fairly limited.

**Comparison $g_4$: $u$ affects only the mean of $p(z \mid u)$**

$$g_4(\tilde{z}, u) = a_1 u + a_2\texttt{sigmoid}(a_3 u + a_4) + a_5\tilde{z} + a_6\texttt{sigmoid}(a_7\tilde{z} + a_8) + a_9 \tag{7}$$

$g_4$ differs from $g_1$ in that the inputs from $u$ are not allowed to modulate the terms that depend on $\tilde{z}$, but that the effect is additive. This means that the parametrization only supports optimal denoising functions for a conditional distribution $p(z \mid u)$ where $u$ only shifts the mean of the distribution of $z$ but otherwise leaves the shape of the distribution intact.

**Results**  All models were tested in a similar setting as the semi-supervised fully connected MNIST task using $N = 1000$ labeled samples. We also reran the best comparison model on $N = 100$ labels. The results of the analyses are presented in Table 5.

As can be seen from the table, the alternative parametrizations of g are inferior to the proposed parametrization at least in the model structure we use.

These results support the finding by Rasmus et al. [7] that modulation of the lateral connection from $\tilde{z}$ to $\hat{z}$ by $u$ is critical for encouraging the development of invariant representation in the higher layers

of the model. Comparison function $g_4$ lacked this modulation and it performed clearly worse than any other denoising function listed in Table 5. Even the linear $g_3$ performed very well as long it had the term $\tilde{z}u$. Leaving the nonlinearity but removing $\tilde{z}u$ in $g_2$ hurt the performance much more.

In addition to the alternative parametrizations for the g-function, we ran experiments using a more standard autoencoder structure. In that structure, we attached an additional decoder to the standard MLP by using one hidden layer as the input to the decoder, and the reconstruction of the clean input as the target. The structure of the decoder was set to be the same as the encoder, that is the number and size of the layers from the input to the hidden layer where the decoder was attached was the same as the number and size of the layers in the decoder. The final activation function in the decoder was set to be the sigmoid nonlinearity. During training, the target was the weighted sum of the reconstruction cost and the classification cost.

We tested the autoencoder structure with 100 and 1000 labeled samples. We ran experiments for all possible decoder lengths, that is we tried attaching the decoder to all hidden layers. However, we did not manage to get significantly better performance than the standard supervised model without any decoder in any of the experiments.

## Footnotes

[4]The whole point of using *denoising* autoencoders rather than regular autoencoders is to prevent skip connections from short-circuiting the decoder and force the decoder to learn meaningful abstractions which help in denoising.

[5]The parametrization can also be interpreted as a miniature MLP network