[Reviews · NeurIPS 2015]

Submitted by Assigned_Reviewer_1

This paper proposes to apply a recent method for deep unsupervised learning called ladder neural network to supervised learning tasks, by combining the original objectives with an additional supervised objective applied at the top of the ladder network. The ladder neural network idea consists of learning as many denoising autoencoding criterions as there are layers in the network, and where the denoising uses the representation at the given layer, and in the next layer.

The method is simple and straightforward, and can be graphically depicted as a neural network (as it is done in Figure 1). Particular attention is dedicated to the choice of the denoising architecture, where the multiplicative interaction between the lateral and top-down connections are made explicit in the model. However, authors show that the choice of denoising model is not crucial, and good results can also be obtained with a variety of denoising models.

Authors demonstrate that the proposed method yields results that are better than the state-of-the-art by a margin on two competitive datasets (MNIST and CIFAR). The improvement is particularly important in the context where few labeled examples are presented compared to the amount of unsupervised data. This shows that the ladder network is particularly suited to capture the class structure in the unlabeled data.

It was not clear to me in the case of the simple Lambda-model, why it is reasonable for practical purposes to only consider the top-level cost in the objective. Robustness to noise in the lower-layer seems to me as important as robustness to noise in the higher layers. Also, it would have been interesting to also consider the complex ladder model (with cost at each layer) in the CIFAR-10 convolutional neural network experiments.
Summary: The paper is an incremental but important extension to a previously proposed method. The paper distinguishes itself by producing state-of-the-art results (by a margin) on several highly competitive datasets.

Submitted by Assigned_Reviewer_2

The paper proposes to extend the work on Ladder networks [1] to incorporate supervised tasks as well. Though the paper achieves good results on semi-supervised learning task in MNIST (in permutation-invariant setting), I have some concerns about the writing and organization of the paper.

Pros: - Neat idea to incorporate in existing feed-forward models - Using unsupervised learning to augment supervised learning is attractive

Cons: - The structure and writing is a concern: -- A lot of details are either postponed to the supplementary material, or are referred to other papers. For example, though Sec2 tries to give us an intuition about Ladder networks, one needs to heavily refer to the original paper [1] in section 3 to understand. -- (typo) L197

- Experiments: -- Their variant of [19] achieves 9.31 error on CIFAR-10, whereas the original paper reports 9.08. The comparison would be fair if authors report the number of [19] on both original-[19] network and their modification, and do the same for the Ladder variant of it. -- The numbers reported [20] for CIFAR-10 are much lower than as originally reported by authors -- The authors mention faster convergence (L391-92), but it is never evaluated. How long does it take for their network to train? How does convergence compare to other methods? Is the convergence speed only because of batch-normalization?

- Related work: In the current work, it seems is just a list of related papers, with few lines about each. Ideally, I would like to see a more coherent related work, and how it relates to the current work.
Summary: The paper proposes to use Ladder networks [1] for the task of Semi-supervised learning. The novelty lies in extending the original work of Valpola [1] to use supervised data/task as well.

Submitted by Assigned_Reviewer_3

A couple comments:

- I think there could be more plots of the intermediate layers in the experiments to expose how they are working.

- Some of the "All" (fully supervised) entries appear missing, in tables 2, 3

- I didn't see the all-layers costs applied in the convnet models, tables 2, 3
Summary: This is a somewhat complex model, with some decent results.

However, in my opinion it could do a better job at illustrating how the model works.

Submitted by Assigned_Reviewer_4

In this paper, the authors proposed a semi-supervised deep learning framework. The proposed method is based on the Ladder network. The authors should provide more discussions and analysis about why the proposed method is better than existing ones. I have several questions on the experiments in this paper:

For the first experiment, why the authors only conduct the experiments on MNIST, not on CIFAR-10?

For Table 2 and Table 3, why the compared methods are different on MNIST and CIFAR-10 datasets?

On the other hand, the reported experimental results are also not as good as the ones achieved by existing methods. For example the results in:

http://rodrigob.github.io/are_we_there_yet/build/classification_datasets_results.html
Summary: The authors proposed a Ladder network based semi-supervised deep learning framework. Utilizing both labeled and unlabeled data in neural network can avoid overfitting and enhance non-linear feature learning. However, the authors didn't provide enough analysis to explain why the proposed model works. The reported experimental results are also not as good as the ones achieved by existing methods.

Submitted by Assigned_Reviewer_5

The manuscript presents impressive semi-supervised results for MNIST using a simple idea related to denoising autoencoders.

The paper proposes a layer-wise reconstruction error of pre-noised activations using a function that is dependent on top down connections and knowledge of the noisy activations at a given layer, i.e. \hat{z}^{l}= g(\noise{z}^{l},\noise{z}^{l+t}).

The approach straightforwardly combines this errors into a scalar loss, combines this unsupervised loss with the supervised loss and trains end-to-end.

the manuscript also presents good, though slightly less striking, results for the CIFAR dataset using the same approach.

It would be nice to see the approach applied to one additional dataset (e.g. demonstrating that it is practical on at least a subset of ImageNet), but I don't believe this is important for acceptance.

While the technique is simple the introduction makes it more difficult to follow than need be.

The didactic Figure 1 (left panel) in particular could be improved for legibility.

Could the choice of the g() function in equation (1) be better justified?

Likely I'm missing something obvious, but why does the vector v have duplicate elements (line 190)?

It may be helpful to other readers to make this clear.

There are also some small miscellaneous grammatical and spelling issues throughout.

The title of the paper might might work better if the work network was pluralized: e.g., "Semi-supervised learning with ladder networks".

When comparing to past results, it should be made clear which past papers made an effort to "balance classes" for the small subsets used for training.

Balancing classes almost certainly helps performance, so it would be good to note which past results had the benefit of balanced classes and which did not.

As a control it would also be nice to compare the results presented with a more "vanilla" semi-supervised denoising autoencoder approach using the same architecture (e.g. batch normalization and relu activations).

This would give us an idea about how much of the performance gain is due to the new ladder-network idea versus just using slightly more updated training methods.

Finally, though the computational costs of the semi-supervised approach seem quite reasonable, but it would be nice to have some measure of the expected wall-clock time/performance for a well optimized version of the ladder network approach versus the purely supervised benchmark.

Demonstration that this kind of semi-supervised approach only requires a modest amount of extra wall-time may help foster wider adoption of this kind of mechanism and help us push the through the tendency to use purely-supervised that currently dominates the field.

Summary: The manuscript presents impressive semi-supervised results for the MNIST and CIFAR datasets using a simple approach related to denoising autoencoders.

However, the presentation of the manuscript could be cleaned up considerably.

Author Feedback
Author rebuttal: Thank you for the feedback and suggestions, we will add clarification where needed and include suggestions as space permits.

We have been improving the paper for a month after the deadline for arXiv, so the writing is in a much better shape for the camera-ready version, but we cannot provide link due to double-blind review.

Reviewers 2, 4, and 7 had questions on (even validity of) our experimental comparisons. Since we claim the state-of-the-art results in many tasks, we hope the reviewers to carefully examine our rebuttal and clarification w.r.t semi-supervised tasks and permutation invariance. These questions can be largely avoided by emphasizing our test setting in the text and table captions.

R4: "For the first experiment, why the authors only conduct the experiments on MNIST, not on CIFAR-10?" and "experimental results are also not as good"
Permutation-invariant MNIST classification is a well-studied problem with a large number of results to compare to. Permutation invariance means that the prior information about the order of pixels is not used in any way (unlike in convolutional modelling or data augmentation) so the results extend to other types of data than images. Note that the 2006 Science paper that started the deep learning boom considered the permutation-invariant MNIST, improving the state of the art from 1.40% to 1.20%. Our improvement is of similar scale. Permutation-invariant CIFAR-10 is not so well studied, so there are no good comparisons. For instance, none of the results in the link you provided is permutation invariant or in semi-supervised setting. To clarify, we improve upon state of the art in the following:
-Permutation-invariant MNIST, all labels, 0.78% -> 0.59%
-Permutation-invariant MNIST, 1000 labels, 2.40% -> 1.05%
-Permutation-invariant MNIST, 100 labels, 3.33% -> 1.30%
-General (convolutional) MNIST, 100 labels, 3.33% -> 0.81%
-CIFAR-10, 500 labels, 46.1% -> 43%
-CIFAR-10, 4000 labels, 31.9% -> 20.7%
We do not claim to improve the fully labeled convolutional case in either MNIST or CIFAR-10.

R2: "The numbers reported [20] for CIFAR-10 are much lower"
Note that Figure 3 of [20] reports results w.r.t. the number of labels per class, whereas we report using number of all labels (so our number is 10x larger). We re-checked the numbers and they were ok.

R2: "[19] achieves 9.31 error -- comparison would be fair if.."
We are not claiming improvement in full-label case, so the purpose of our 9.27 number is to tell that we were successfully able to replicate architecture ConvPool-CNN-C [19] that we chose as our baseline. The reason we did not choose ALL-CNN-C is that it performed worse with 500 and 4000 labels, even though it was better with full labels, see L287-290. So we chose the better performing and fairer model to compare against in semi-supervised setting.

R4: "For Table 2&3, why the compared methods are different on MNIST and CIFAR-10?"
If you mean the methods in the literature, we used those semi-supervised results that we could find. If you have suggestions, we will gladly add more comparison results. If you mean why we used a smaller network for MNIST than CIFAR-10 ourselves, it was because the dataset is much simpler and lower-dimensional (28x28x1 vs. 32x32x3).

R1&7: "Results for convolutional Ladder network with all examples are missing from tables 2, 3":
We left the cells blank simply because we focused on semi-supervised tasks and did not perform the experiments at all. The column "All" in Table 2 exists only to demonstrate that we were able to reproduce the result in [19] and in Table 3 to inform what is the performance of our baseline model. We should've mentioned this explicitly to avoid confusion and will fix this.

R2: "The structure and writing is a concern"
We agree. This has been addressed in the arXiv version which has much cleaner structure and writing, including improved section on related work.

R1&7: "Why wasn't the full Ladder tested in ConvNet experiments? tables 2, 3"
We ran out of time before the deadline so we left it as future work but we have some initial results in our arXiv version.

R2: "authors mention faster convergence (L391-92)"
We did not intend to claim it is faster than other methods; the text says "convergence is fast", because on MNIST the baseline takes 1 hour, Gamma 1.5 hours, and full Ladder 2 hours on GPU. We will clarify this and avoid claiming anything on something we cannot support by experiments.

R1: "extension to a previously proposed method"
Note that this would be the first conference paper on Ladder networks, there is only a book chapter and arXiv/workshop presentations.

R7: "there could be more plots of the intermediate layers -- to expose how they are working"
We will look into that.

R8, Thank you for the good suggestions, many of your points have already been improved in the arXiv version. We will pay attention to the class balancing.